# Affiliate Stigma and Related Factors in Family Caregivers of Children with Attention-Deficit/Hyperactivity Disorder

**DOI:** 10.3390/ijerph17020576

**Published:** 2020-01-16

**Authors:** Chih-Cheng Chang, Yu-Min Chen, Tai-Ling Liu, Ray C. Hsiao, Wen-Jiun Chou, Cheng-Fang Yen

**Affiliations:** 1Department of Psychiatry, Chi Mei Medical Center, Tainan 70246, Taiwan; rabiata@gmail.com; 2Department of Health Psychology, College of Health Sciences, Chang Jung Christian University, Tainan 71101, Taiwan; 3Department of Psychiatry, Kaohsiung Medical University Hospital, Kaohsiung 80708, Taiwan; bluepooh79@msn.com (Y.-M.C.); dai32155@gmail.com (T.-L.L.); 4Department of Psychiatry, School of Medicine, and Graduate Institute of Medicine, Kaohsiung Medical University, Kaohsiung 80708, Taiwan; 5Department of Psychiatry and Behavioral Sciences, University of Washington School of Medicine, Seattle, WA 98195-6560, USA; rhsiao@u.washington.edu; 6Department of Psychiatry, Children’s Hospital and Regional Medical Center, Seattle, WA 98105, USA; 7College of Medicine, Chang Gung University, Taoyuan 33302, Taiwan; 8Department of Child and Adolescent Psychiatry, Chang Gung Memorial Hospital, Kaohsiung Medical Center, Kaohsiung 83301, Taiwan

**Keywords:** attention-deficit/hyperactivity disorder, caregivers, affiliate stigma

## Abstract

This cross-sectional questionnaire study examined factors related to affiliate stigma among caregivers of children with attention-deficit/hyperactivity disorder (ADHD) and the association of affiliate stigma with caregivers’ unfavorable attitude toward ADHD and moderators. The affiliate stigma of 400 caregivers of children with ADHD was assessed using the Affiliate Stigma Scale. Caregivers’ and children’s factors related to affiliate stigma were examined using multiple regression analysis. Associations of affiliate stigma with caregivers’ unfavorable attitudes toward children’s diagnoses, pharmacotherapy, behavioral therapy, and biological explanations of the etiologies of ADHD were examined using logistic regression analysis. Female caregivers and those caring for girls with ADHD had higher levels of affiliate stigma than did male caregivers and those caring for boys. Higher education levels in caregivers and more severe inattention symptoms in children were associated with higher levels of affiliate stigma. A higher level of affiliate stigma was also significantly associated with unfavorable attitudes toward children’s diagnoses, pharmacotherapy and behavioral therapy, and etiological explanations for ADHD. Multiple factors of caregivers and children were related to affiliate stigma in caregivers of children with ADHD. Affiliate stigma is significantly associated with caregivers’ unfavorable attitude toward ADHD.

## 1. Introduction

### 1.1. Stigma to Attention-Deficit/Hyperactivity Disorder

Attention-deficit/hyperactivity disorder (ADHD) is a prevalent neurodevelopmental disorder with a lifetime prevalence of 10.1% according to the Diagnostic and Statistical Manual of Mental Disorders, Fifth Edition (DSM-5) diagnostic criteria [1] in a nationally representative sample of children in Taiwan [2]. Although neurocognitive etiologies contribute to ADHD symptom development [1], ADHD symptoms are often attributed to affected children’s unwillingness in self-control or intentional opposition and misunderstanding of ADHD increases both social rejection and hostile emotions in peers, teachers, family members, and neighbors [3]. The National Stigma Study-Children revealed that stigmatized attitudes toward children with ADHD are prevalent among American adults [4]. However, ADHD-related stigma and its negative influence on affected children and families in Asian countries are underexplored.

Studies have examined ADHD-related public stigma, self-stigma, and courtesy stigma and their negative effects. A high level of perceived public stigma toward ADHD can predict low willingness to receive intervention [5] and low medication adherence in children with ADHD [6]. Children with ADHD may internalize public stigma to develop self-stigma [7], which may compromise their self-esteem and emotional regulation [8]. Courtesy stigma is negative judgment toward family members or people close to a stigmatized person due to their relationship with the stigmatized target [9]. A review of parents of children with ADHD indicated that fighting a crossfire of blame, self-blame, and stigmatization is a major challenge of living with a child with ADHD [10,11]. Many parents not only are concerned with how society labels, isolates, and rejects their children but also worry about the possible effects of diagnosis and treatment on their children’s self-esteem and opportunities for future success [12]. Parental stigma levels are also negatively related to parents’ and children’s willingness to use community health programs [13], highlighting the need for community outreach and public health programs that address and eliminate ADHD-related public, self, and courtesy stigma [12].

### 1.2. Affiliate Stigma in Caregivers of People with Mental Illness

Affiliate stigma in caregivers of people with mental illness develops through perceiving and internalizing public stigma toward caregivers [14]. Caregivers with intense affiliate stigma may agree with public stigma toward them (cognitive component); feel shame, embarrassment, and negative emotions stemming from internalized stigma (affective component); and withdraw from social relations or alienate themselves from closely affiliated stigmatized family members or people (behavioral component) [14,15]. Affiliate stigma may result in more serious consequences than courtesy stigma does [16]. Affiliate stigma may not only enhance caregivers’ psychological distress and reduce their quality of life [17,18] but also abate the care they provide for affiliated people and cooperation with health care professionals [19]. It is crucial to evaluate and reduce affiliate stigma among caregivers of people with mental illness [20].

### 1.3. Affiliate Stigma in Caregivers of Children with ADHD

A growing body of research has investigated affiliate stigma in family caregivers of people with schizophrenia, depression, or affective disorders [19]. These studies have not only demonstrated that affiliate stigma is prevalent among family caregivers but also revealed that health-focused interventions for caregivers can reduce the effect of affiliate stigma by providing social support from medical professionals [19]. However, few studies have focused on affiliate stigma in family caregivers of children with ADHD. A study on parents of children with ADHD in the United States demonstrated that greater affiliate stigma was associated with more negative parenting, children’s poorer social skills, and greater aggression [21]. In a study on French mothers of children with ADHD, affiliate stigma was positively associated with mothers’ distress and children’s ADHD symptoms [22]. In most cases, ADHD is first diagnosed in childhood because of educational, emotional, and social adjustment problems caused by the core symptoms of inattention and hyperactivity–impulsivity [5]. Because family caregivers might best understand their children with ADHD and can help health care professionals effectively assist them, affiliate stigma in family caregivers of children with ADHD warrants further study to provide evidence for developing intervention programs to reduce stigma.

### 1.4. Aims of the Present Study

Cultural background affects levels of stigmatization toward ADHD [23]. People in Taiwan are collectivistic orientated and emphasize social relationships and harmony [24]. Children with ADHD may be labelled as disruptors of order in schools and neighborhoods and thus may be stigmatized. Moreover, Taiwanese people are highly influenced by Confucianism and value academic achievement. Thus, children with ADHD may be blamed for difficulties in academic performance and regarded as “bad students” by school staff. Because family caregivers of children with ADHD may be blamed for neglecting their duty to care for their children, they may always face the threat of “losing face” [25]. Therefore, the risk of affiliate stigma increases.

Rather than focusing on courtesy stigma addressing public views toward caregivers, the present study focused on affiliate stigma in caregivers of children with ADHD. Based on sociocultural background, this cross-sectional questionnaire study examined the level of affiliate stigma, its related sociodemographic and illness factors, and its association with unfavorable attitudes toward ADHD and moderators among family caregivers of children with ADHD in Taiwan. Identification of factors that contribute to developing affiliate stigma is crucial in promoting interventions and prevention. A review of studies demonstrated that results are mixed regarding associations between affiliate stigma and sociodemographic characteristics (such as age, gender, and education levels) of caregivers and affiliated people with mental illnesses (mainly schizophrenia and affective disorders) [19]. In addition, meta-analyses revealed that no sociodemographic or symptom severity factor was significantly associated with affiliate stigma [19]. To our knowledge, only one study examined the relationship between children’s ADHD symptoms and affiliate stigma [22], whereas no study has examined the sociodemographic factors of affiliate stigma in family caregivers of children with ADHD. On the basis of studies on sociodemographic factors related to courtesy stigma in ADHD [23] and the relationship between affiliate stigma and children’s ADHD symptoms [22], we hypothesized that sociodemographic and ADHD symptom factors are related to affiliate stigma among family caregivers of children with ADHD in Taiwan.

The present study also examined the association of affiliate stigma with unfavorable attitude toward the diagnosis, treatment, and etiological explanation of ADHD and moderators among family caregivers of children with ADHD in Taiwan. Although neurocognitive etiologies and treatment for ADHD have been proposed [26], caregivers often have an ambivalent view of evaluation, diagnosis, and intervention results, and this negatively affects treatment continuity in children with ADHD [27]. Perceived public stigma negatively influences parental attitudes toward children’ diagnosis, symptom etiology, and the necessity and continuity of treatment for ADHD [28,29,30,31]. Whether affiliate stigma is similarly associated with unfavorable attitude toward the diagnosis, treatment, and etiological explanation of ADHD among family caregivers of children with ADHD in Taiwan warrants further study. Whether sociodemographic factors and ADHD symptoms moderate this association also requires examination. We hypothesized that affiliate stigma is associated with family caregivers’ unfavorable attitude toward children’s diagnoses, treatment, and etiological explanation of ADHD and that sociodemographic and ADHD symptom characteristics have moderating effects on these associations.

## 2. Methods

### 2.1. Participants and Procedure

Caregivers of children who were aged 18 years or younger and had received a diagnosis of ADHD according to DSM-5 criteria [1] were consecutively recruited for this study between June 2018 and April 2019 from the child and adolescent psychiatric outpatient clinics of two medical centers in Kaohsiung, Taiwan. Two child psychiatrists conducted diagnostic interviews with children and caregivers and established ADHD diagnoses based on DSM-5 criteria. Multiple data sources—including clinical observation of each child’s behavior and caregivers’ ratings of ADHD symptoms on the short version of the Swanson, Nolan, and Pelham, Version IV, Scale (SNAP-IV), Chinese version [32,33]—were used to support each diagnosis. Children who had an intellectual disability or autism spectrum disorder with difficulties in communication were excluded. Caregivers who had an intellectual disability, schizophrenia, bipolar disorder, or any cognitive deficits that resulted in significant communication difficulties were also excluded. A total of 409 caregivers of children who received an ADHD diagnosis were invited to participate in the study. Of these, nine (2.2%) declined to participate. Thus, 400 (97.8%) caregivers participated in the study and were interviewed by research assistants by using a research questionnaire (Figure 1). The Institutional Review Boards (IRBs) of Kaohsiung Medical University (KMUHIRB-E(I)-20180179) and Chang Gung Memorial Hospital, Kaohsiung Medical Center (201800723A3) approved the study.

### 2.2. Measures

#### 2.2.1. Affiliate Stigma Scale

The Affiliate Stigma Scale (ASS) is a self-rated 22-item questionnaire measuring caregivers’ internalization of stigma toward family members’ mental illness [14]. In accordance with the aim of the present study, we focused on caregivers’ affiliate stigma toward their children’s ADHD. The ASS included three domains: affect (7 items; for example, “I feel inferior because one of my children has ADHD.”), cognition (7 items; for example, “My reputation is damaged because I have a child with ADHD at home.”), and behavior (8 items; for example, “I dare not tell others that I have a child with ADHD”). Each item of the ASS asks respondents to rate their agreement from 1 (strongly disagree) to 4 (strongly agree) on a 4-point Likert scale. A higher score on the ASS indicates that the caregiver has a higher level of self-stigma toward their child’s ADHD. The original version exhibited excellent internal consistency (α = 0.94) and satisfactory predictive validity [14]. The ASS also has robust psychometric properties in a Taiwanese sample [20]. In the present study, Cronbach’s α values for the affect, cognition, and behavior domains and the total ASS were 0.88, 0.89, 0.89, and 0.95, respectively.

#### 2.2.2. Caregivers’ Attitudes toward Children’s ADHD

We invited the caregivers to answer four items on the research questionnaire to evaluate caregivers’ attitudes toward their children’s diagnoses, pharmacotherapy and behavioral therapy, and biological explanation of ADHD etiology on a 4-point scale. Given that the participants were recruited from outpatient clinics and were supposed to have a higher understanding of ADHD than caregivers of children with ADHD who did not visit outpatient clinics for ADHD problems, the present study classified caregivers who rated items with 1 (strongly unacceptable), 2 (mildly unacceptable), and 3 (mildly acceptable) as having unfavorable and those who rated items with 4 (strongly acceptable) as having favorable attitudes, respectively, toward diagnosis, pharmacotherapy, psychotherapy, and etiological explanations of their children’s ADHD.

#### 2.2.3. Chinese Version of the SNAP-IV Scale, Parent Form

The short, Chinese version of SNAP-IV was used to assess caregiver-reported severity of ADHD symptoms exhibited in the preceding month. This version comprises 26 items encompassing the core DSM-derived ADHD subscales of inattention, hyperactivity and impulsivity, and oppositional symptoms of oppositional defiant disorder [32,33]. Each item is rated on a 4-point Likert scale ranging from 0 (not at all) to 3 (very much). In the present study, Cronbach’s α values for inattention, hyperactivity and impulsivity, and oppositional behavior were 0.89, 0.90, and 0.92, respectively.

#### 2.2.4. Caregivers’ and Children’s Factors

Caregivers completed a questionnaire for collecting their sex (female or male), age (years), years of education completed, marital status (married, divorced, or separated), occupational socioeconomic status (SES), and frequency of attending religious activities (frequently, occasionally, or never). The caregivers could request assistance with any problems they encountered in completing the questionnaire. According to marriage status, participants were grouped into those who were married and those who were divorced or separated. Occupational SES was assessed using the Close-Ended Questionnaire of the Occupational Survey (CEQ-OS) [34], which classifies paternal and maternal occupational SES into five levels such that a higher level indicates high occupational SES. The CEQ-OS has acceptable reliability and validity and has been used frequently in studies on children and adolescents in Taiwan [34]. Participants whose CEQ-OS was level I, II, or III were classified as low occupational SESs, and those whose CEQ-OS was level IV or V were classified as high occupational SESs [35]. Caregivers’ frequency of attending religious activities were classified into high (frequently) and low (occasionally or never). The research assistants conducted an interview with children with ADHD to collect their sex (girl or boy), age (years), and education level (kindergarten, primary school, or high school).

### 2.3. Statistical Analysis

Data analysis was performed using SPSS 24.0 (SPSS Inc., Chicago, IL, USA). Affiliate stigma levels on the three domains of the ASS were calculated. Caregivers’ and children’s factors related to affiliate stigma on the ASS were examined using multiple regression analysis. A two-tailed *p* value of less than 0.05 was considered significant. The associations of caregiver affiliate stigma with unfavorable attitudes toward children’s diagnoses, pharmacotherapy and behavioral therapy, and biological etiologies of ADHD were examined using logistic regression analysis by controlling caregivers’ and children’s factors. Odds ratios (ORs) and 95% confidence intervals (CIs) were used to indicate significance. We also used criteria proposed by Baron and Kenny [36] to examine moderators in the association between affiliate stigma and caregivers’ unfavorable attitudes toward ADHD.

### 2.4. Ethics

The study procedures were performed in accordance with the Declaration of Helsinki. The IRBs of Kaohsiung Medical University Hospital and Chang Gung Memorial Hospital, Kaohsiung Medical Center approved the study. All participants were briefed on the study and provided written informed consent before completing research questionnaires.

## 3. Results

Table 1 presents caregivers’ and children’s demographic characteristics, affiliate stigma, unfavorable attitude toward children’s ADHD, and children’s ADHD symptoms. The ASS score in the affect domain was significantly higher than those in the cognition (paired *t* = 23.22, *p* < 0.001) and behavior domains (paired *t* = 22.88, *p* < 0.001). The ASS score in the cognition domain was significantly higher than that in the behavior domain (paired *t* = 4.14, *p* < 0.001). Scores in the three ASS domains were highly correlated with each other (Pearson’s correlation *r* = 0.679–0.869). Therefore, the total ASS score was used to represent the level of affiliate stigma in caregivers of children with ADHD in further statistical analyses.

Table 2 presents the multiple regression analysis results of caregivers’ and children’s factors related to affiliate stigma on the ASS. In Model I, female caregivers had a higher level of affiliate stigma on the ASS than male caregivers did. Higher education level among caregivers was significantly associated with a higher level of affiliate stigma. In Model II, caregivers of girls with ADHD had a higher level of affiliate stigma than those of boys with ADHD did. Inattention symptom severity in children was positively associated with the level of affiliate stigma. 

Table 3 displays logistic regression analysis results for the association between affiliate stigma and caregivers’ unfavorable attitudes toward children’s ADHD. After controlling for the effects of caregivers’ and children’s factors, a higher level of affiliate stigma based on the total ASS score was significantly associated with unfavorable attitudes toward children’s diagnoses, pharmacological and behavioral therapy, and etiological explanations of ADHD.

Because caregivers’ education level and children’s age were significantly associated with unfavorable attitude toward ADHD diagnosis, the moderating effects of caregivers’ education level and children’s age on the association between affiliate stigma and unfavorable attitudes toward ADHD diagnosis were examined. The interaction between affiliate stigma and caregiver’s education level (OR = 1.062, 95% CI: 0.900–1.254) and between affiliate stigma and children’s age (OR = 1.011, 95% CI: 0.889–1.150) were not associated with unfavorable attitude toward ADHD diagnosis. The results did not support the moderating effect of caregiver’s education level and children’s age.

## 4. Discussion

In this study, although the three domains of affiliate stigma were highly correlated with each other, affiliate stigma in the affect domain was significantly higher than that in the cognition and behavior domains. Female sex and higher education level for caregivers and female sex and more severe inattention symptoms in children were significantly associated with higher levels of affiliate stigma. Affiliate stigma increased with caregivers’ unfavorable attitudes toward children’s diagnoses, pharmacological and behavioral therapy, and etiological explanations of ADHD. Sociodemographic factors and ADHD symptoms did not moderate the association between affiliate stigma and caregivers’ unfavorable attitudes toward ADHD.

### 4.1. Components of Affiliate Stigma

Affiliate stigma consists of three interrelated components: cognition, affect, and behavioral responses [15]. Because affiliate stigma is the process and result of agreeing with public attitudes toward caregivers because of internalized stigma, the cognition component is the core of affiliate stigma. Caregivers may then react affectively by feeling shameful and embarrassed and react behaviorally by withdrawing from social relations or alienating themselves from targeted individuals to avoid association [14,15,37,38]. In this study, the level of affective affiliate stigma surpassed those of cognitive and behavioral affiliate stigma. Caregivers of children with ADHD may feel that their caregiving burden is intolerable, which may lead to negative emotions [22]. Parental depression may compromise parental self-efficacy [39] and parent–child interaction quality [40,41]. The results of the present study indicated that family caregivers of children with ADHD have various domains of affiliate stigma. Although these various domains of affiliate stigma are related, each domain may require a unique program for intervention. For example, caregivers with high affective affiliate stigma may require sufficient and skillful emotional support to relieve their feelings of shame and embarrassment due to internalized stigma.

### 4.2. Factors Related to Affiliate Stigma

In this study, female sex and education level were positively associated with affiliate stigma in caregivers. Studies on caregivers of people with schizophrenia, affective disorders, or intellectual disabilities did not observe a significant association of caregivers’ sex and education level with affiliate stigma [14,18,42,43,44]. In traditional Taiwanese families, mothers are expected to care for children [45]. Mothers of children with ADHD may be more frequently blamed for neglecting their parental duty to instruct children to behave than fathers are. Mothers of children with ADHD may also have more opportunities than fathers to interact with parents of other children, who are a main source of stigma toward children with ADHD [46]. Caregivers with higher education levels may have higher expectations for children’s success in academic performance and daily practice, as well as expectations from themselves and others to parent successfully. These unmet expectations due to children’s ADHD may increase affiliate stigma in female and highly educated caregivers.

Affiliated children’s female sex and inattention symptom severity were significantly associated with higher affiliate stigma levels. Results of video studies on the sex difference in stigma toward ADHD are mixed. Pescosolido et al. revealed that participants were more likely to avoid male children with ADHD [47], whereas Fausett demonstrated that negative peer ratings were more likely if the ADHD-associated deviant behavior was displayed by a fictitious female character [48]. Traditional Chinese concepts of gender roles and stereotypes still heavily influence Taiwanese society. For example, traditional Chinese societies may rate a typical male as higher on extroversion and a typical female higher on restraint [49]. Shyness is even considered a characteristic of girls in traditional Chinese societies [50]. Traditional Chinese concepts on gender roles may aggravate social prejudice against girls with ADHD who do not conform to gender stereotypes and exacerbate affiliate stigma in family caregivers of girls with ADHD.

Charbonnier et al. demonstrated that children’s ADHD symptoms were positively associated with affiliate stigma, which was positively associated with mothers’ distress [22]. However, the study summarized children’s inattention and hyperactivity–impulsivity symptoms for analysis [22], whereas the present study revealed that children’s inattention but not hyperactivity–impulsivity symptoms were significantly associated with affiliate stigma. For children, inattention symptoms may increase difficulty in studies, thus deepening affiliate stigma when caregivers face teachers’ and family members’ censures for failing to monitor their children’s academic achievements. Parents with affiliate stigma may have an urge to correct their children’s ADHD symptoms because they feel ashamed and perceive their children’s symptoms as negatively affecting how others view them as parents [21]. The results of this study indicated that those developing psychosocial interventions in affiliate stigma among caregivers of children with ADHD should consider caregivers’ sex and education level and children’s sex and inattention symptoms and make adequate adjustments accordingly.

### 4.3. Association of Affiliate Stigma with Unfavorable Attitude toward ADHD

Affiliate stigma was positively associated with caregivers’ unfavorable attitudes toward children’s diagnoses, pharmacological and behavioral therapy, and etiological explanations of ADHD. A study also demonstrated that parental attitudes toward ADHD treatment were associated with susceptibility to ADHD stigma, ADHD knowledge, and misconceptions [51]. Family caregivers may use various strategies to manage self-stigma and public stigma, including denying a child’s ADHD diagnosis and refusing to cooperate with health care professionals [13]. These coping strategies may result from caregivers’ intention to reduce public accusations of not fulfilling their parental role [13]. 

### 4.4. Limitations

The present study has several limitations. First, the cross-sectional design limited the possibility of determining a causal relationship of affiliate stigma with unfavorable attitude toward ADHD. In addition to the contribution of affiliate stigma to developing unfavorable attitudes toward ADHD, unfavorable attitude toward pharmacotherapy and behavioral therapy for ADHD may delay opportunities for treatment for children with ADHD and worsen their inattention symptoms, thus deepening caregivers’ affiliate stigma. Second, data for both affiliate stigma and children’s ADHD symptoms were reported by caregivers. The single data source may have resulted in common-method variance. Third, the present study did not examine the roles of multidimensional psychosocial factors, such as social support or health belief, in affiliate stigma or the association between affiliate stigma and caregivers’ unfavorable attitudes toward ADHD.

### 4.5. Implications

Based on the results of the present study, we recommend that health care professionals view affiliate stigma as a critical topic that warrants vigorous evaluation and intervention. Affiliate stigma in family caregivers of children with ADHD results from interactions among multiple ecological systems. Children with ADHD and their caregivers may benefit from “ecologically sensitive” treatment in which children, caregivers, social environments, and broader political and cultural contexts that shape children’s behaviors are carefully investigated [52]. In particular, affiliate stigma may be shaped by family caregivers’ exposure to negative media and blame from educational systems. Therefore, policies that mitigate distorted images spread in the media of ADHD and adequate communication strategies between education systems and caregivers may have meaningful implications for reducing the development of affiliate stigma. Because family caregivers with affiliate stigma may not spontaneously seek clinical assistance for their children with ADHD, health care professionals can perform outreach in communities to provide psychoeducation [21]. Caregivers’ subjective experiences should be emphasized and carefully examined to identify their needs [53]. Caregivers with high affiliate stigma may also benefit from interacting with those with low affiliate stigma; in the interaction process, they may develop mutual understanding and support that help reduce affiliate stigma [53]. Moreover, behavioral parent training to reduce affiliate stigma should be provided to family caregivers of children with ADHD with adequate adjustments for caregivers’ and children’s sociodemographic characteristics and ADHD symptoms. Family supportive groups may provide caregivers with emotional support to reduce care burdens.

## 5. Conclusions

Multiple factors of caregivers and children were related to affiliate stigma in caregivers of children with ADHD. Prevention and intervention programs for reducing affiliate stigma in caregivers of children with ADHD should take these related factors into consideration. Moreover, affiliate stigma is significantly associated with caregivers’ unfavorable attitude toward children’s diagnoses, pharmacotherapy and behavioral therapy, and etiological explanations for ADHD. The result supported that reducing affiliate stigma is important for treatment in ADHD.

## Figures and Tables

**Figure 1 ijerph-17-00576-f001:**
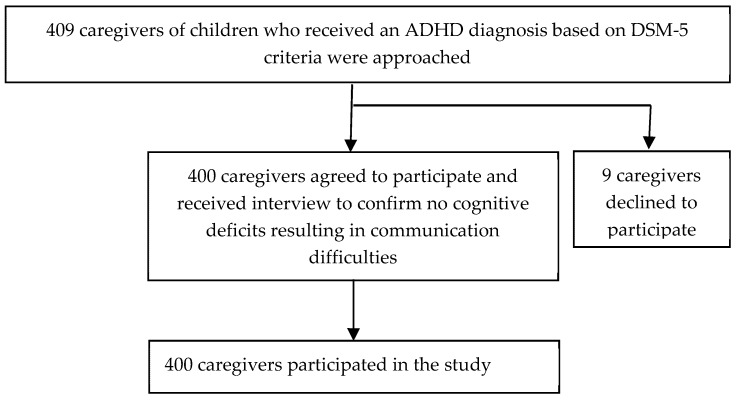
Flowchart of study design. ADHD: attention-deficit/hyperactivity disorder; DSM-5: Diagnostic and Statistical Manual of Mental Disorders, Fifth Edition.

**Table 1 ijerph-17-00576-t001:** Caregivers’ and children’s demographic characteristics, mean score on each domain of affiliate stigma, unfavorable attitude toward children’s attention-deficit/hyperactivity disorder (ADHD), and children’s ADHD symptoms (*N* = 400).

Variables	*n* (%)	Mean (SD)	Range
Caregivers			
Relationship with the child			
Mother	287 (71.8)		
Father	90 (22.5)		
Others	23 (5.8)		
Age (years)		43.4 (6.8)	25–70
Sex			
Female	304 (76.0)		
Male	96 (24.0)		
Education (years)		13.8 (2.9)	3–23
Parental marriage status			
Married	320 (80)		
Divorced or separated	80 (20)		
Occupational socioeconomic status			
High	155 (38.8)		
Low	245 (61.2)		
Frequency of attending religious activities			
High	143 (35.8)		
Low	257 (64.3)		
Level of affiliate stigma			
Affect		2.1 (0.7)	1–4
Cognition		1.6 (0.5)	1–3.4
Behavior		1.6 (0.5)	1–3.5
Total		1.8 (0.5)	1–3.4
Unfavorable attitude toward children’s ADHD			
Diagnosis	173 (43.3)		
Pharmacotherapy	180 (45)		
Behavioral therapy	112 (28)		
Biological explanation for etiologies	108 (27)		
Children			
Age (years)		10.7 (3.2)	4–18
Sex			
Girls	64 (16.0)		
Boys	336 (84.0)		
Education			
Primary school or kindergarten	355 (88.8)		
High school	45 (11.3)		
ADHD symptoms on the SNAP-IV			
Inattention		13.4 (3.6)	0–27
Hyperactivity/impulsivity		9.8 (6.0)	0–27
Opposition defiance		10.1 (6.0)	0–24

ADHD: attention-deficit/hyperactivity disorder; SNAP-IV: Swanson, Nolan, and Pelham, Version IV Scale.

**Table 2 ijerph-17-00576-t002:** Caregivers’ and children’s factors related to total affiliate stigma.

Variables	Model I			Model II		
	Beta	*t*	*p*	Beta	*t*	*p*
Caregivers factors						
Sex (0: female; 1: male)	−0.102	−1.987	0.048	−0.073	−1.468	0.143
Age	0.053	1.015	0.311	0.085	1.624	0.105
Education level	0.136	2.457	0.014	0.112	2.102	0.036
Marriage status (0: married; 1: divorced or separated)	−0.029	−0.572	0.568	−0.040	−0.830	0.407
Frequency of attending religious activities (0: high; 1: low)	0.088	1.744	0.082	0.087	1.799	0.073
Occupational socioeconomic status (0: high; 1: low)	0.084	1.541	0.124	0.034	0.643	0.521
Children’s factors						
Sex (0: girl; 1: boy)				−0.122	−2.522	0.012
Age				0.052	0.964	0.336
Inattention				0.253	4.284	<0.001
Hyperactivity/impulsivity				0.030	0.451	0.652
Opposition defiance				0.027	0.423	0.672

**Table 3 ijerph-17-00576-t003:** Associations of sociodemographic variables and affiliate stigma with caregivers’ unfavorable attitude toward children’s ADHD.

Variables	Diagnosis			Pharmacotherapy			Behavioral Therapy			Biological Explanation for Etiologies		
	Wals χ^2^	*p*	OR (95% CI)	Wals χ^2^	*p*	OR (95% CI)	Wals χ^2^	*p*	OR (95% CI)	Wals χ^2^	*p*	OR (95% CI)
Caregivers’ sex	1.667	0.197	0.707 (0.418–1.197)	0.056	0.814	1.062 (0.644–1.753)	2.893	0.089	1.614 (0.930–2.803)	0.006	0.936	0.977 (0.545–1.751)
Caregivers’ age	1.420	0.233	0.979 (0.946–1.014)	2.112	0.146	0.975 (0.943–1.009)	3.614	0.057	0.963 (0.927–1.001)	3.254	0.071	0.964 (0.927–1.003)
Parental education level	6.160	0.013	0.900 (0.828–0.978)	0.938	0.333	0.962 (0.889–1.041)	1.250	0.263	0.950 (0.868–1.039)	1.815	0.178	0.939 (0.856–1.029)
Caregivers’ marriage status	0.016	0.899	0.966 (0.564–1.653)	0.280	0.597	0.868 (0.515–1.464)	0.694	0.405	0.771 (0.418–1.422)	1.432	0.232	0.680 (0.361–1.279)
Caregivers’ frequency of religious activities	0.008	0.930	0.980 (0.625–1.537)	0.022	0.883	1.033 (0.668–1.597)	0.346	0.557	1.161 (0.706–1.908)	1.295	0.255	1.345 (0.807–2.241)
Caregivers’ socioeconomic status	0.227	0.634	1.124 (0.695–1.818)	0.105	0.746	0.926 (0.580–1.477)	0.202	0.653	1.129 (0.665–1.917)	0.180	0.671	1.124 (0.656–1.925)
Children’s sex	0.135	0.713	1.115 (0.625–1.989)	0.074	0.785	0.924 (0.524–1.631)	0.059	0.808	1.080 (0.579–2.016)	0.103	0.748	0.904 (0.488–1.676)
Children’s age	5.071	0.024	0.917 (0.850–0.989)	1.469	0.226	0.956 (0.889–1.028)	0.988	0.320	1.042 (0.961–1.129)	0.009	0.924	1.004 (0.925–1.090)
Inattention	0.547	0.460	0.982 (0.936–1.031)	0.014	0.905	1.003 (0.957–1.051)	0.394	0.530	0.983 (0.933–1.036)	0.491	0.484	0.981 (0.930–1.035)
Hyperactivity/impulsivity	3.689	0.055	0.952 (0.905–1.001)	0.217	0.642	0.989 (0.942–1.037)	1.339	0.247	0.968 (0.916–1.023)	0.386	0.534	1.018 (0.963–1.076)
Opposition defiance	0.000	0.985	1.000 (0.955–1.049)	0.055	0.814	0.995 (0.950–1.041)	0.877	0.349	0.976 (0.927–1.027)	3.161	0.075	0.953 (0.904–1.005)
Affiliate stigma	31.984	<0.001	3.888 (2.429–6.225)	21.410	<0.001	2.862 (1.833–4.468)	26.033	<0.001	3.667 (2.226–6.040)	25.932	<0.001	3.800 (2.273–6.352)

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
