# Peer review of "Affiliate Stigma and Related Factors in Family Caregivers of Children with Attention-Deficit/Hyperactivity Disorder"

_ijerph, 2020, doi:10.3390/ijerph17020576_

Round 1
Reviewer 1 Report
The stigma and isolation of families with a child with a disease or disability is a topic of social interest for a better quality of life for children and their families. In the case of symptoms presented by the child, it leads to disruptive behavior or with outsourcing
behaviors become more evident. This work delves into this situation.
Below are some comments / questions and suggestions, personally:
1. I do not understand the reason why the objective of the article is included in a section that focuses on stigma in caregivers of people with mental illness (67-68). I believe that the objective should go to the end of the introduction section. And in the introduction there are words in bold and larger font, I guess it is a typo.
2. When the sample is described, a better description of it would be clarifying by means of a flow.
3. In the methodology I think it would be convenient to specify how sociodemographic aspects are evaluated and how information was collected. It would be advisable to explain the reason why the attitude test towards treatment and diagnosis is considered 1-3 as an unfavorable attitude.
4. I think it is better to specify that the attitude towards treatment and diagnosis of children with ADHD is evaluated, sometimes it can create confusion (165 or 225).
5. With regard to sociodemographic variables, the classification regarding parental marital status is strange, and it is not specified how occupational socioeconomic status or religious activities have been categorized
6. Regarding the results, the affiliate stigma levels are low, less than 2, less in affection than it is 2.1. The intervals include decimals, I understand that it is the average results of the items, perhaps it should be explained in the legend of the table.
7. Sometimes causal statements are made, when the study presented does not allow it (221-222)
8. If the table 2 is placed horizontally, I think it helps to read it. And the title of this table does not reflect that sociodemographic variables are analyzed.
Author Response
Comment
I do not understand the reason why the objective of the article is included in a section that focuses on stigma in caregivers of people with mental illness (67-68). I believe that the objective should go to the end of the introduction section.
Response
Thank you for your suggestion. We moved the objective of the article into “1.4. Aims of the present study.” Please refer to line 103-104.
Comment
And in the introduction there are words in bold and larger font, I guess it is a typo. Please refer to line 62.
Response
Thank you for your reminding. We corrected the typo in the revised manuscript.
Comment
When the sample is described, a better description of it would be clarifying by means of a flow.
Response
We added a flowchart to illustrate the sample. Please refer to line 152-156.
Comment
In the methodology I think it would be convenient to specify how sociodemographic aspects are evaluated and how information was collected.
Response
Thank you for your suggestion. In the revised manuscript we revised the contents to introduce how sociodemographic aspects are evaluated and how information was collected as below. Please refer to line 190-192 and line 202-204.
“Caregivers completed a questionnaire for collecting their sex (female or male), age (years), years of education completed, marital status (married, divorced, or separated), occupational socioeconomic status (SES) and frequency of attending religious activities (frequently, occasionally or never). The caregivers could request assistance with any problems they encountered in completing the questionnaire…The research assistants conducted an interview with children with ADHD to collect their sex (girl or boy), age (years), and education level (kindergarten, primary school, or high school).”
Comment
It would be advisable to explain the is considered 1-3 as an unfavorable attitude.
Response
We added explanation for the method to clarify the participants according to their levels of attitudes towards diagnoses, treatment, and etiology as below. Please refer to line 174-179.
“Given that the participants were recruited from outpatient clinics and were supposed to have a higher understanding of ADHD than caregivers of children with ADHD who did not visit outpatient clinics for ADHD problems, the present study classified caregivers who rated items with 1 (strongly unacceptable), 2 (mildly unacceptable) and 3 (mildly acceptable) as having unfavorable and those who rated items with 4 (strongly acceptable) as having favorable attitudes, respectively,…”
Comment
I think it is better to specify that the attitude towards treatment and diagnosis of children with ADHD is evaluated, sometimes it can create confusion (165 or 225).
Response
We revised the contents to introduce how caregivers’ attitudes towards children’s ADHD diagnoses, treatment, and etiology were collected as below. Please refer to line 172-174.
“We invited the caregivers to answer four items on the research questionnaire to evaluate caregivers’ attitudes toward their children’s diagnoses, pharmacotherapy and behavioral therapy, and biological explanation of ADHD etiology on a 4-point scale.”
Comment
With regard to sociodemographic variables, the classification regarding parental marital status is strange, and it is not specified how occupational socioeconomic status or religious activities have been categorized
Response
We revised the introduction for the classification of the participants according to marital status, occupational socioeconomic status, and religious activities as below. Please refer to line 190-195 and line 199-204. Parental marriage status in Table 1 and Table were also revised into “married vs. divorced or separated” instead of “intact vs. disrupted.”
“According to marriage status, participants were grouped into those who were married and those who were divorced or separated.” “Participants whose CEQ-OS was level I, II, or III were classified as low occupational SESs, and those whose CEQ-OS was level IV or V were classified as high occupational SESs.” “Caregivers’ frequency of attending religious activities were classified into high (frequently) and low (occasionally or never).”
Comment
Regarding the results, the affiliate stigma levels are low, less than 2, less in affection than it is 2.1. The intervals include decimals, I understand that it is the average results of the items, perhaps it should be explained in the legend of the table.
Response
Thank you for your reminding. We added “mean score on each domain of affiliate stigma” into the title of Table 1. Please refer to line 230.
Comment
Sometimes causal statements are made, when the study presented does not allow it (221-222).
Response
Thank you for your reminding. We revised the description as below to replace causal statements. Please refer to line 238-239.
“Inattention symptom severity in children was positively associated with the level of affiliate stigma.?
Comment
If the table 3 is placed horizontally, I think it helps to read it. And the title of this table does not reflect that sociodemographic variables are analyzed.
Response
In the revised manuscript we placed table 3 horizontally. We also revised the title to include sociodemographic variables. Please refer to line 253.
Reviewer 2 Report
Thank you for your work in this area of ADHD. The article was well-written and flowed nicely in English. You also addressed many of the areas I thought of as I was reading through this in your discussion and adequately (in my opinion) discussed future directions, implications, clinical relevance, and weaknesses of the study. You did a very nice job outlining why you chose to examine affiliate stigma in Taiwan. I had not been previously familiar with the term affiliate stigma, but you clearly explained what it is in your paper.
One factor I was curious about was looking at the relationship between affiliate stigma and inattention symptoms when controlling for child gender. Studies have demonstrated that females are more likely than males to present with inattentive symptoms, and your study established the relationship between higher affiliate stigma and the female child gender. I am wondering if the relationship you found between inattentive symptoms and affiliate stigma still exists when accounting for child gender.
I would also like to see a little more explanation for your choice of unfavorable and favorable caregiver attitudes towards treatment, diagnoses, etc, with your splitting of items rated 1-3 as unfavorable and as 4 as favorable. I see the anchors of 1 being "strongly unacceptable" to 4 being "strongly acceptable," but what is unclear to me is what 3 is stated as. There would be a difference, to me, if it was "acceptable" or something worded as being "unacceptable" like "mildly unacceptable." In the first case (if it said 3 = "acceptable" or even "mildly acceptable"), I would think that a 3 would be a favorable rating and not unfavorable. However, if a 3 was worded as "unacceptable" to whatever extent, then I can see why the choice was made to keep a 3 rating as unfavorable.
Author Response
Comment
One factor I was curious about was looking at the relationship between affiliate stigma and inattention symptoms when controlling for child gender. Studies have demonstrated that females are more likely than males to present with inattentive symptoms, and your study established the relationship between higher affiliate stigma and the female child gender. I am wondering if the relationship you found between inattentive symptoms and affiliate stigma still exists when accounting for child gender.
Response
Thank you for your comment. In Model II, both the sex of girl (p =.012) and inattention symptom severity (p <.001) in children were positively associated with the level of affiliate stigma, indicating that both being the caregivers of girls and inattention symptom severity in children were independently associated with affiliate stigma in caregivers.
Comment
I would also like to see a little more explanation for your choice of unfavorable and favorable caregiver attitudes towards treatment, diagnoses, etc, with your splitting of items rated 1-3 as unfavorable and as 4 as favorable. I see the anchors of 1 being "strongly unacceptable" to 4 being "strongly acceptable," but what is unclear to me is what 3 is stated as. There would be a difference, to me, if it was "acceptable" or something worded as being "unacceptable" like "mildly unacceptable." In the first case (if it said 3 = "acceptable" or even "mildly acceptable"), I would think that a 3 would be a favorable rating and not unfavorable. However, if a 3 was worded as "unacceptable" to whatever extent, then I can see why the choice was made to keep a 3 rating as unfavorable.
Response
Thank you for your comment. We added explanation for the method to clarify the participants according to their levels of attitudes towards diagnoses, treatment, and etiology as below. Please refer to line 174-179.
“Given that the participants were recruited from outpatient clinics and were supposed to have a higher understanding of ADHD than caregivers of children with ADHD who did not visit outpatient clinics for ADHD problems, the present study classified caregivers who rated items with 1 (strongly unacceptable), 2 (mildly unacceptable) and 3 (mildly acceptable) as having unfavorable and those who rated items with 4 (strongly acceptable) as having favorable attitudes, respectively,…”